# Polyhydroxybutyrate Production from Methane and Carbon Dioxide by a Syntrophic Consortium of Methanotrophs with Oxygenic Photogranules without an External Oxygen Supply

**DOI:** 10.3390/microorganisms11051110

**Published:** 2023-04-24

**Authors:** Selim Ashoor, Seong-Hoon Jun, Han Do Ko, Jinwon Lee, Jérôme Hamelin, Kim Milferstedt, Jeong-Geol Na

**Affiliations:** 1Department of Agricultural Microbiology, Faculty of Agriculture, Ain Shams University, Hadayek Shoubra, Cairo 11241, Egypt; selim_ashoor@agr.asu.edu.eg; 2Department of Chemical and Biomolecular Engineering, Sogang University, Seoul 04107, Republic of Korea; cleversam10@gmail.com (S.-H.J.); dodo3612@hanmail.net (H.D.K.); jinwonlee@sogang.ac.kr (J.L.); 3INRAE, University of Montpellier, LBE, 102 Avenue des Etangs, 11100 Narbonne, France; jerome.hamelin@inrae.fr (J.H.); kim.milferstedt@inrae.fr (K.M.)

**Keywords:** greenhouse gas mitigation, polyhydroxybutyrate, syntrophic process, methanotrophy, oxygenic photogranules

## Abstract

Here, a syntrophic process was developed to produce polyhydroxy-β-butyrate (PHB) from a gas stream containing CH_4_ and CO_2_ without an external oxygen supply using a combination of methanotrophs with the community of oxygenic photogranules (OPGs). The co-culture features of *Methylomonas* sp. DH-1 and *Methylosinus trichosporium* OB3b were evaluated under carbon-rich and carbon-lean conditions. The critical role of O_2_ in the syntrophy was confirmed through the sequencing of 16S rRNA gene fragments. Based on their carbon consumption rates and the adaptation to a poor environment, *M. trichosporium* OB3b with OPGs was selected for methane conversion and PHB production. Nitrogen limitation stimulated PHB accumulation in the methanotroph but hindered the growth of the syntrophic consortium. At 2.9 mM of the nitrogen source, 1.13 g/L of biomass and 83.0 mg/L of PHB could be obtained from simulated biogas. These results demonstrate that syntrophy has the potential to convert greenhouse gases into valuable products efficiently.

## 1. Introduction

The mitigation of greenhouse gases is essential for developing a carbon-neutral society. Among these gases, methane (CH_4_) is particularly significant due to its high heat-trapping potential, with a global warming potential around 80 times greater than CO_2_ in the short term [1]. CH_4_ emissions account for 18% of the total GHG emissions, even if it is emitted into the atmosphere in much smaller quantities than CO_2_ [2]. This high impact led to international initiatives such as the Global Methane Pledge. Consequently, the development of efficient processes to convert CH_4_ into value-added products is crucial.

Biological conversion is one potential approach to CH_4_ mitigation [3]. Biological routes for CH_4_ conversion offer the advantage of transforming CH_4_ into valuable substances under mild physicochemical conditions, unlike thermochemical routes, which are capital- and energy-intensive processes [4]. Methanotrophs, crucial players in the global methane cycle, are utilized as catalysts for this process. These bacteria are classified into type I, type II, and type X based on their physiological characteristics and methane assimilation pathways. The unique metabolic pathway of methanotrophs involves the initial step of methane oxidation, resulting in methanol production. Subsequently, methanol is further oxidized to formaldehyde, which is assimilated to form intermediates of the central metabolism, leading to the synthesis of various compounds [5].

Several bioproducts have been reported from CH_4_ conversion using methanotrophs, including hydrogen [6], alcohols [7,8], organic acids [9], amino acids [10], lipids [11], single-cell proteins [12,13], and biodegradable plastics [14,15,16]. Among these bioproducts, the production of biodegradable plastics using methanotrophs has garnered attention as it has the potential to address both plastic pollution and climate change, two of humanity’s most pressing challenges. Type Ⅱ methanotrophs can store excess carbon as polyhydroxy-β-butyrate (PHB), a representative biodegradable polymer, when certain growth-essential elements, such as nitrogen, sulfur, and phosphorous, are limited [17,18]. Previous research has indicated that nitrogen deficiency is the most effective method for inducing the accumulation of PHB in type Ⅱ methanotrophs, including *M. trichosporium* OB3b [19]. 

However, there are challenges to the full-scale implementation of biological CH_4_ conversion for biodegradable polymers, including inherent properties of the gaseous substrates required for methanotrophs, i.e., CH_4_ and O_2_ [20]. Gas transfer attributes are known to dictate overall productivity; the solubility of these gases amounts to only 19 mg/L for water in contact with a 100% CH_4_ headspace and 38 mg/L for water in contact with a 100% O_2_ headspace under typical growth conditions at 30 °C and 1 atm [21,22]. The solubility issue is further exacerbated by the dilution of nitrogen in the air for the two gaseous substrates. Additionally, safety concerns arise due to the flammability range of CH_4_ explosions in the air [13], which is between 5.5 and 14%. While anaerobic CH_4_ oxidation routes may address some of these drawbacks, the slow growth of the corresponding microorganisms presents obstacles to the practical application [23].

Recently, there have been attempts to simulate the division of labor strategy, which has been employed by microbial species to solve natural difficulties for various applications [24]. One such application is the syntrophic culture of methanotrophs and phototrophs [25,26,27,28], in which the O_2_ required for methanotrophs is supplied through photosynthesis. This approach could address mass transfer limitations and safety issues and transform CH_4_ and CO_2_ simultaneously. While there are some reports on syntrophic cultivations of methanotrophs and phototrophs, most of these studies have focused on wastewater treatment or greenhouse gas conversion rather than on specific bioproducts such as biodegradable polymers.

Here, we demonstrate a syntrophic process of methanotrophs with phototrophs in oxygenic photogranules (OPGs) to produce PHB from a gas stream containing CH_4_ and CO_2_. OPGs consist of phototrophic cyanobacteria interacting syntrophically with heterotrophic bacteria and have been recently described in the context of wastewater treatment [28]. It may be possible to convert this syntrophy into a community where methanotrophs mainly substitute the heterotrophic partners. This syntrophic community may then be able to transform methane into metabolites of interest. Furthermore, we anticipated that the nitrogen limitation in the culture medium under excess carbon may enhance PHB accumulation by the consortium of methanotrophs with OPGs.

Prior to assessing the production of PHB through syntrophy, the properties of co-cultures of type I and type II methanotrophs were evaluated concerning their gas intake profile and population dynamics to select the most suitable type of methanotroph for the syntrophic process. Subsequently, a series of experiments were conducted to improve PHB production in the syntrophy by adjusting the nitrogen source concentration. To the best of our knowledge, this is the first investigation into the production of PHB through biological CH_4_ conversion without an external O_2_ supply.

## 2. Materials and Methods

### 2.1. Microorganisms and Culture Conditions

*Methylosinus trichosporium* OB3b (NCIMB 11131) and *Methylomonas* sp. DH-1 strains were cultivated in 500 mL baffled flasks (polycarbonate) with a butyl-rubber septum containing 100 mL of nitrate mineral salts (NMS) medium, which was composed of KNO_3_ (1 g), MgSO_4_·7H_2_O (1 g), CaCl_2_·2H_2_O (0.2 g), CuSO_4_·5H_2_O (0.0025 g), Fe-EDTA (0.0038 g), Na_2_MoO_4_·2H_2_O (0.0006 g), 1 mL trace element solution (EDTA (250 mg), FeSO_4_·7H_2_O (500 mg), ZnSO_4_·7H_2_O (400 mg), CoCl_2_·6H_2_O (50 mg), MnCl_2_·7H_2_O (20 mg), H_3_BO_3_ (15 mg), NiCl_2_·6H_2_O (10 mg), HCl (30 mL) per liter), 10 mL phosphate solution (Na_2_HPO_4_·7H_2_O (62 g), KH_2_PO_4_ (26 g) per liter), and 1 mL vitamin stock (calcium pantothenate (50 mg), nicotinamide (50 mg), thiamine HCl (50 mg), riboflavin (50 mg), folic acid (20 mg), biotin (20 mg), and vitamin B_12_ (1 mg) per liter) [29]. All components of the medium were added before sterilization, except phosphate and vitamin solutions. The sterilization was performed for 15 min at 121 °C. Phosphate and vitamin solutions were sterilized separately and then added to the medium. A gas mixture of 30% (*v/v*) CH_4_ and 70% (*v/v*) air was supplied by a mass flow controller (IMC1300, ISVT Co., Ltd., Yongin-si, Republic of Korea). The headspace gas was replenished every two days. The flasks were incubated in a shaking incubator at 230 rpm and 30 °C.

The oxygenic photogranules (OPGs) were cultivated in 1 L glass baffled flasks sealed with a butyl-rubber septum using 500 mL of simplified synthetic wastewater (SW) medium, which consisted of 139.45 mg CH_3_COONa, 60.02 mg NH_4_Cl, 27.64 mg urea, 12.00 mg KH_2_PO_4_, 22.4 mg NaHPO_4_·7H_2_O, 20.6 mg MgSO_4_·7H_2_O, and 2.90 mg FeSO_4_·7H_2_O per liter. Prior to sterilization, all medium components were added except urea, KH_2_PO_4_, and FeSO_4_·7H_2_O, which were separately sterilized and then added to the medium. Carbon dioxide and air were supplied with a ratio of 15% (*v/v*) CO_2_ and 85% (*v/v*) air by headspace gas substitution with a mass flow controller (IMC1300, ISVT Co., Ltd., Yongin-si, Republic of Korea). The headspace gas was replenished once every five days. The flasks were incubated on an orbital shaker at room temperature and 100 rpm under LED at a light intensity of 145 μmol/m^2^/s.

### 2.2. Gas Consumption of the Syntrophic Consortiums

Each strain of methanotrophic bacteria was co-cultured with OPGs separately in 60 mL serum bottles sealed with a butyl-rubber septum. The working volume in each serum bottle was 12 mL of mixed media (NMS medium and SW medium at a ratio of 1:1). All media components were added before autoclaving except urea, KH_2_PO_4_, FeSO_4_·7H_2_O, phosphate, and vitamin solutions, which were added after autoclaving. The inoculation ratio of methanotrophic bacteria and OPGs was 1:1 (0.1%: 0.1% (wet *w/v*)). Carbon-rich conditions (30% CH_4_ and 20% CO_2_ with N_2_ balance) and carbon-lean conditions (6% CH_4_ and 4% CO_2_ with N_2_ balance) were employed with both consortiums. Gases were supplied by headspace gas substitution using a mass flow controller (IMC1300, ISVT Co., Ltd., Yongin-si, Republic of Korea). The serum bottles were incubated on an orbital shaker at room temperature and 100 rpm, maintaining a constant light intensity of 145 μmol/m^2^/s. All experiments were performed in duplicate. The gas composition in the headspace was measured periodically to determine gas intake. Samples were taken after cultivation to check the microbial community using 16S rRNA amplicon sequencing.

### 2.3. 16S rRNA Amplicon Sequencing 

DNA was extracted using a DNeasy PowerSoil Kit (Qiagen, Germany) and quantified by Quant-IT PicoGreen (Invitrogen, Carlsbad, CA, USA). The Illumina standard protocols were used to amplify the 16S rRNA gene (V3 and V4 regions) for preparing the sequencing libraries. PCR amplification was performed using 2 ng of the extracted DNA, 500 nM of each universal primer (F/R), 0.5 U of Herculase II fusion DNA polymerase (Agilent Technologies, Santa Clara, CA, USA), 1 mM of dNTP mix, and 5x reaction buffer. The cycle conditions for 1st PCR were 95 °C for 3 min, then 25 cycles of 95 °C (0.5 min), 55 °C (0.5 min), and 72 °C (0.5 min), with a final extension for 5 min (72 °C). The following forward and reverse primers with Illumina overhang adapter sequences were employed for 1st PCR amplification:

V3-F (5′-TCGTCGGCAGCGTCAGATGTGTATAAGAGACAGCCTACGGGNGGCWGCAG-3′)

V4-R (5′-GTCTCGTGGGCTCGGAGATGTGTATAAGAGACAGGACTACHVGGGTATCTAATCC-3′)

AMPure beads (Agencourt Bioscience, La Jolla, CA, USA) were used to purify the 1st PCR product. After that, 2nd PCR was carried out using 2 μL of 1st PCR purified product and NexteraXT Indexed Primer to construct the final library. The 2nd PCR cycle conditions were analogous to those of the 1st PCR cycle, except that 10 cycles were performed instead of 25. The purification of 2nd PCR product was conducted with AMPure beads, and the quantification was carried out using qPCR following the protocol guide (KAPA Library Quantification kits for Illumina platforms). TapeStation D1000 ScreenTape (Agilent Technologies, USA) was used to qualify the 2nd PCR product. Sequencing of paired-end (2 × 300 bp) was conducted by Macrogen using the MiSeq™ platform (Illumina, San Diego, CA, USA). QIIME (version 1.9.0) was employed to analyze the sequencing data [30]. BLASTn (version 2.4.0) was used to align the sequence of each operational taxonomic unit (OTU) sequence against the NCBI 16S rRNA database reference sequences to obtain the taxonomic affiliation of each OTU. 

### 2.4. Flask Cultivation of Syntrophic Consortium for PHB Production Using CH_4_ and CO_2_


*Methylosinus trichosporium* OB3b and OPGs were co-cultured in 500 mL polycarbonate baffled flasks sealed with a butyl-rubber septum. Various concentrations of N-NO_3_ and N-NH_4_ as nitrogen sources (11.5 mM, 5.7 mM, 2.9 mM, and 1.4 mM) were employed to investigate the effect of nitrogen concentration. The working volume of each flask was 100 mL of media based on the NMS and SW medium at 1:1, except for the nitrogen source. All media components were added before sterilization except urea, KH_2_PO_4_, FeSO_4_·7H_2_O, phosphate, and vitamin solutions, which were added after sterilization. The inoculation ratio of methanotrophic bacteria and OPGs was 1:1 (0.1%: 0.1% (wet *w/v*)). The mixture comprising 30% CH_4,_ 20% CO_2,_ and 50% N_2_ was supplied by headspace gas substitution using a mass flow controller (IMC1300, ISVT Co., Ltd., Yongin-si, Republic of Korea). The headspace gas was replenished every two days. All flasks were incubated on an orbital shaker at room temperature and 100 rpm, maintaining a constant light intensity of 145 μmol/m^2^/s. The total incubation time was 7 days. All experiments were conducted in duplicate. The gas composition in the headspace was measured every two days before the gas exchange. Total dry cell biomass and PHB production were determined at the end of the experiments. 

### 2.5. Analytical Methods

The headspace gases were analyzed using a gas chromatograph (YL6500 GC, YOUNG IN Chromass, Anyang-si, Republic of Korea) equipped with 80/100 Porapak N and 45/60 Molecular Sieve 13X columns (Supelco Inc., Bellefonte, PA, USA) and a thermal conductivity detector (TCD). The carrier gas was argon (15 mL/min flow rate). The temperatures of the oven, injector, and detector were 40 °C, 120 °C, and 120 °C, respectively. All analyses were performed in duplicate.

PHB contents were analyzed using the GC-FID method [31,32]. PHB was extracted from the dried cell pellet samples by the solvent extraction method using methanol and chloroform and then trans-esterified with an acid catalyst [32]. The amount of PHB was determined using a gas chromatograph (YL6500 GC, YOUNG IN Chromass, Anyang-si, Republic of Korea) equipped with a J&W DB-WAX 123-7033 GC column from Agilent Technologies and a flame ionization detector (FID) using benzoic acid as an internal standard. The carrier gas was helium (3 mL/min flow rate). The injector and detector were maintained at 280 °C and 300 °C, respectively. The oven temperature was initially maintained at 85 °C for 5 min and then increased gradually to 200 °C. All analyses were carried out in duplicate.

## 3. Results and Discussion

### 3.1. Gas Consumption by the Syntrophic Consortiums under Different CH_4_ Contents

To verify the implementation of the syntrophic system of methanotrophs and phototrophs for CH_4_ conversion, we cultivated the consortium with a CH_4_/CO_2_ gas mixture in the absence of externally supplied O_2_. Two different conditions were tested, differing in the carbon content of the gas mixture: a carbon-rich condition (50% carbon) and a carbon-lean condition (5% carbon). Both conditions had a CH_4_ to CO_2_ ratio of 3:2, representing a typical biogas composition [33]. In addition to CH_4_ and CO_2_, the gas mixture contained only N_2_ as an inert gas. 

Under the carbon-rich condition, both *Methylomonas* sp. DH-1 and *M. trichosporium* OB3b strains were cultivated with OPGs to achieve simultaneous consumption of the CH_4_/CO_2_ mixture. During the first few days of cultivation, CH_4_ consumption was higher than CO_2_ consumption in both cases, even though the aqueous solubility of CO_2_ is known to be higher than that of CH_4_ [34], which can be attributed to the higher growth rate of methanotrophs compared to phototrophs in OPGs, resulting in an imbalance between CO_2_ production by methanotrophs and CO_2_ consumption by phototrophs, and thus a significant decrease in the proportion of CO_2_ in the headspace was not observed. After that, the growth rate of phototrophs increased, resulting in concurrent consumption of the CH_4_/CO_2_ mixture (Figure 1a,c). Significantly, the proportion of CH_4_ and CO_2_ in the headspace was observed to be below 5%, suggesting that the co-culture strategy can achieve complete abatement of both greenhouse gases. Comparing the performances of the two consortiums, the consortium of *M. trichosporium* OB3b with OPGs showed a higher carbon consumption rate than that of *Methylomonas* sp. DH-1 with OPGs. Despite the reports that type I methanotrophs (namely, *Methylomonas* sp. DH-1) exhibit superior growth rates generally [17], it is worth noting that the consortium using *M. trichosporium* OB3b, a type II methanotroph, is better at consuming the gaseous substrate. These results may be due to the inferior adaptability of the type Ⅰ methanotrophs, such as *Methylomonas* sp. DH-1, to grow with low concentrations of O_2_ produced in situ by phototrophic cyanobacteria. Type Ⅰ methanotrophs prefer relatively low CH_4_ and high O_2_ concentrations, while type Ⅱ methanotrophs, including *M. trichosporium* OB3b, favor the opposite [35]. The accumulation of O_2_ and the consumption of CO_2_ during cultivation suggest that photosynthetic cyanobacteria were more active than methanotrophic bacteria in the consortium. In the control condition with only OPGs (Figure 1e), CH_4_ consumption was negligible, indicating that the CH_4_ in the system was assimilated by the added *M. trichosporium* OB3b or *Methylomonas* sp. DH-1. It was also observed that CO_2_ was consumed more rapidly in the co-culture with OB3b than in the culture using only OPGs (Figure 1c,e). Taking into account the production of CO_2_ from CH_4_ metabolism, the difference in performances is likely even more significant, highlighting the synergistic effect of the syntrophic conversion of greenhouse gases. Despite OPGs being initially inoculated as granules, methanotrophs and OPGs grew individually suspended without integration in the co-cultivation, implying a potential shift in the composition of the syntrophic populations.

Under the carbon-lean condition, using the *M. trichosporium* OB3b strain, the syntrophic consortium consumed the CH_4_/CO_2_ mixture (Figure 1d). However, the consortium using the *Methylomonas* sp. DH-1 strain displayed low efficacy (Figure 1b). While both CH_4_ and CO_2_ were consumed upon the initial gas injection, the subsequent gas exchange resulted in decreased activity of the methanotrophs, leading to insufficient CH_4_ consumption. After examining the pH of the final samples, it was determined that the consortium with *Methylomonas* sp. DH-1 and *M. trichosporium* OB3b had pH of 6.95 and 6.79, respectively, under carbon-rich conditions. On the other hand, under carbon-lean conditions, their respective pH was 8.60 and 7.22. The pH increase may be attributed to the low concentration of CO_2_ and/or the unequal cellular uptake of cations and anions under these conditions. The higher pH observed with the consortium with *Methylomonas* sp. DH-1 impeded the activity of the methanotroph. Most methanotrophic bacteria are neutrophilic microorganisms, as methane oxidation occurs optimally under neutral conditions [29].

Repeated batch cultivations were carried out under the carbon-lean conditions by replenishing the headspace of the culture with a CH_4_/CO_2_ mixture whenever the CH_4_ was entirely consumed. The conversion of CH_4_ and CO_2_ consistently decreased with increasing batches, suggesting that the syntrophic populations changed differently under these conditions compared to the carbon-rich condition.

### 3.2. Metagenome Sequencing of the Syntrophic Consortiums after Cultivation with Different Gas Ratios

We sequenced 16S rRNA amplicons of syntrophic consortia cultivated at different carbon availabilities and using the two types of methanotrophs. The dominant bacterial species present in OPGs were *Sphingomonas echinoides*, *Stanieria cyanosphaera*, *Bradyrhizobium namibiense,* and *Bosea vaviloviae,* accounting for 51.0%, 25.6%, 10.3%, and 5.8%, respectively (Figure 2). The proportion of heterotrophs in OPGs decreased when co-cultured with the methanotrophs. This effect was more pronounced in the presence of *Methylomonas* sp. DH-1, a type I methanotroph. In the consortium containing *Methylomonas* sp. DH-1, the methanotroph was the most abundant sequence in the amplicon, with a relative abundance of 52.0%. The second most abundant strain was *Stanieria cyanosphaera,* with a relative abundance of 33.8%. *Stanieria cyanosphaera* is a unicellular and spherical cyanobacterium with cell sizes ranging from 5 to 40 μm [36]. Finding unicellular cyanobacteria in OPGs is unusual since previous studies mostly found filamentous cyanobacteria as keystone species for OPG [28,37] and it explains the reason for the suspension culture of OPGs and methanotrophs with the disintegration of the granules. Other less abundant species, such as *Pseudomonas veronii*, *Pseudoxanthomonas mexicana,* and *Pelomonas puraquae,* were found, accounting for 3.4%, 3.0%, and 2.7%, respectively. In the consortium containing *M. trichosporium* OB3b, *Stanieria cyanosphaera* was the most dominant strain, accounting for 65.1%. A higher proportion of *Stanieria cyanosphaera* may have regularly supplied enough O_2_ to *M. trichosporium* OB3b, leading to a more efficient CH_4_ metabolism. This may explain the faster CH_4_ consumption of *M. trichosporium* OB3b compared to *Methylomonas* sp. DH-1 under carbon-rich conditions. The relative abundance of *M. trichosporium* OB3b was 9.0%. Other bacteria in this consortium included *Flavihumibacter cheonanensis*, *Altererythrobacter aurantiacus,* and *Caedimonas varicaedens*, with a relative abundance of 14.2%, 3.6%, and 2.7%, respectively. Under the carbon-lean condition, the relative abundance of *Stanieria cyanosphaera* decreased to 42.7%, while *M. trichosporium* OB3b increased to 20.9%. The proportion of *Altererythrobacter aurantiacus* and *Caedimonas varicaedens* also increased to 10.1% and 9.7%, respectively. *Flavihumibacter cheonanensis* was not detected under this condition. These findings suggest that the ratio of cyanobacteria to the remaining other bacteria in the consortium is lower under the carbon-lean condition, which may contribute to decreased gas ingestion. The O_2_ supply plays a crucial role in the consortium’s performance under all conditions. Given that the syntrophic system relies on photosynthetic reactions to O_2_ production, the *M. trichosporium* OB3b strain, which is robust against carbon availability and O_2_ concentrations, may be more suitable as a workhorse for this system. In addition, *M. trichosporium* OB3b can synthesize PHB, a biodegradable polymer derived from the acetyl-CoA pool of the serine cycle [38].

### 3.3. Improvement of PHB Accumulation by Adjusting Nitrogen Source Concentrations

We cultivated a syntrophic consortium comprising *M. trichosporium* OB3b and phototrophs in OPGs without an external O_2_ supply at various nitrogen source concentrations in the culture media to investigate the effect of nitrogen limitation on PHB accumulation. Under 11.5 mM nitrogen (100% of NMS and SM medium), the O_2_ concentration in the flask headspace significantly increased (Figure 3a) while the CO_2_ concentration decreased. These changes in concentrations may be attributed to the higher phototrophic activity of OPGs under this condition, as they were producing O_2_ from CO_2_ at a higher rate than its consumption by *M. trichosporium* OB3b. These results are consistent with those of Rasouli et al. [39], who observed an accumulation of 27% O_2_ at the end of co-culturing *Chlorella sorokiniana* and *Methylococcus capsulatus*. The high level of CH_4_ consumption under these conditions also suggests the active growth of *M. trichosporium* OB3b, which can utilize CH_4_ as a sole energy and carbon source [5,40] in conjunction with the abundant O_2_ produced by OPGs.

As nitrogen source concentration in the culture medium decreased, the O_2_ concentration in the flask headspace decreased, while the CO_2_ concentration increased (Figure 3b–d). It may be due to the reduction in the phototrophic activity of OPGs rather than the higher growth of *M. trichosporium* OB3b, as CH_4_ consumption also decreased.

Biomass and PHB concentrations of the syntrophic consortium are presented in Table 1. The maximum biomass (1.30 g/L) was achieved using 11.5 mM nitrogen in the culture medium, with 33.3 mg/L of PHB concentration and 2.6 mg/100 mg of PHB content, respectively. While biomass production was almost comparable at 1.29 g/L with 5.7 mM nitrogen, PHB production (36.6 mg/l) and content (2.8 mg/100 mg biomass) increased. At 2.9 mM nitrogen, biomass concentration decreased to 1.13 g/L, but PHB concentration and content increased to 83.0 mg/L and 7.4 mg/100 mg biomass, respectively. At the lowest nitrogen concentration of 1.4 mM, PHB concentration decreased to 67.3 mg/L even if PHB content increased to 13.7 mg/100 mg biomass, resulting from a significant reduction in total biomass (0.50 g/L). It is likely due to the nitrogen limitation in the culture medium, which stimulates PHB accumulation within *M. trichosporium* OB3b cells but hinders the growth of the syntrophic consortium. Overall, the best PHB accumulation by the syntrophic consortium in this study was achieved at a nitrogen concentration of 2.9 mM in the culture medium. PHB production and content were lower than pure methanotrophic cultivations. Despite similar methane consumption rates in the syntrophic culture and pure culture of *M. trichosporium* OB3b (Figure 3c,e), PHB productivity in the co-culture was inferior. This lower performance can be attributed to the higher prevalence of oxygen-producing OPGs relative to PHB-producing methanotrophs within the consortium. Thus, further optimization of microbial populations may be necessary to improve PHB synthesis in co-culture conditions.

## 4. Conclusions

In this study, we explored applying a syntrophic community using methanotrophs and phototrophs in OPGs to address the challenges in the biological conversion of CH_4_ concerning economic feasibility and effective utilization of gaseous substrates. It was found that the steady supply of O_2_ is a crucial factor in the performance of the consortium. Type II methanotrophs that are less sensitive to carbon availability and O_2_ levels are a more suitable partner for co-cultivation with photosynthetic bacteria. We confirmed the production of biodegradable polymers from greenhouse gases using the consortium under nitrogen limitation in the culture medium. Additionally, the syntrophic relationship between methanotrophs and phototrophs, involving the exchange of CO_2_ and O_2_, allows for the complete removal of CO_2_ generated by CH_4_ consumption, potentially leading to a zero-carbon emission process. These results demonstrate the potential to transform greenhouse gases, comprising CH_4_ and CO_2_, into biodegradable polymers through syntrophy without an external O_2_ supply. However, issues such as O_2_ accumulation or the remaining CH_4_ due to an imbalance between CH_4_ and CO_2_ consumption must be addressed. Methanotrophs and OPGs grew individually suspended without integration in the co-cultivation. Notably, 16S rRNA amplicon sequencing revealed that unicellular cyanobacteria dominate over filamentous cyanobacteria, which play an essential role in granulation, explaining this suspension growth. For practical applications such as cell immobilization and light transmission, methanotrophs should be fully incorporated into the OPGs to create single methanotrophic photogranules. Further research on optimizing parameters in the culture environment, including controlling populations in the consortium and designing co-culture media, is necessary to address these limitations.

## Figures and Tables

**Figure 1 microorganisms-11-01110-f001:**
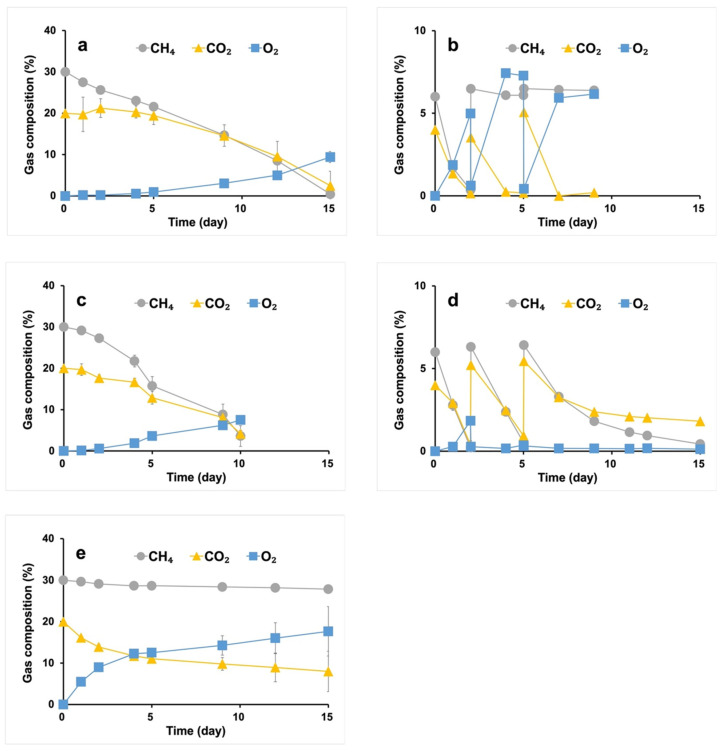
Gas uptake profile of syntrophic system of *Methlyomonas* sp. DH-1 with 50% (*v/v*) N_2_ (**a**) and 90% (*v/v*) N_2_ (**b**), *M. trichosporium* OB3b with 50% (*v/v*) N_2_ (**c**) and 90% (*v/v*) N_2_ (**d**), and OPGs (control) with 50% (*v/v*) N_2_ (**e**). Excluding N_2_, the only gas species are CH_4_ and CO_2_, and the ratio of CH_4_ to CO_2_ is 3:2.

**Figure 2 microorganisms-11-01110-f002:**
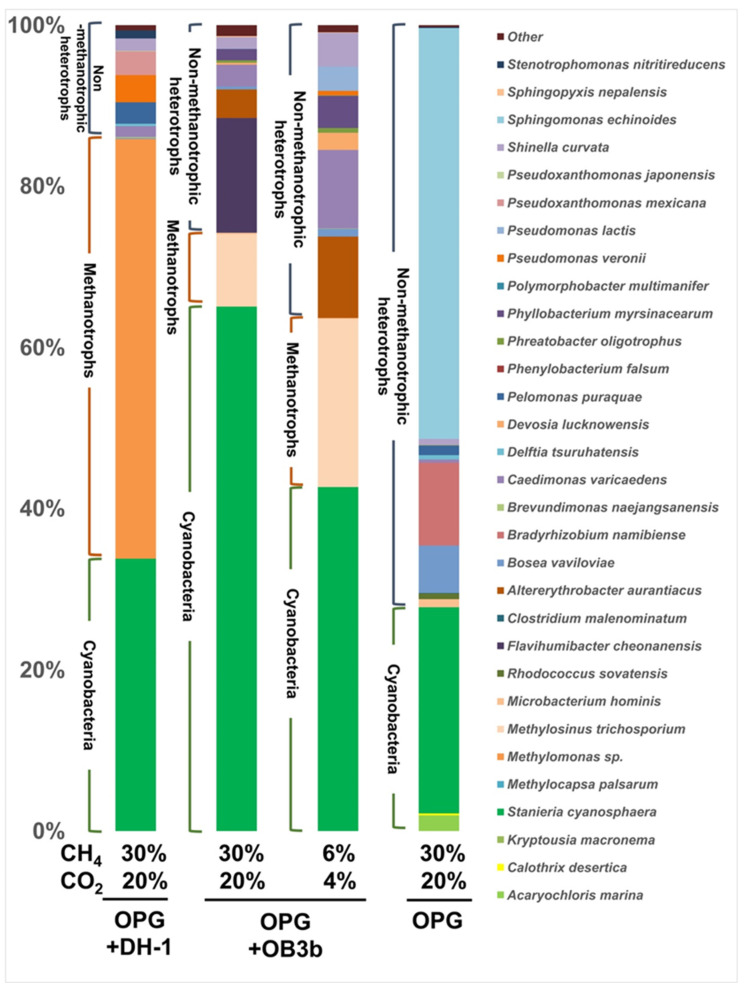
Microbial community composition of the syntrophic consortiums after co-culturing with different gas ratios.

**Figure 3 microorganisms-11-01110-f003:**
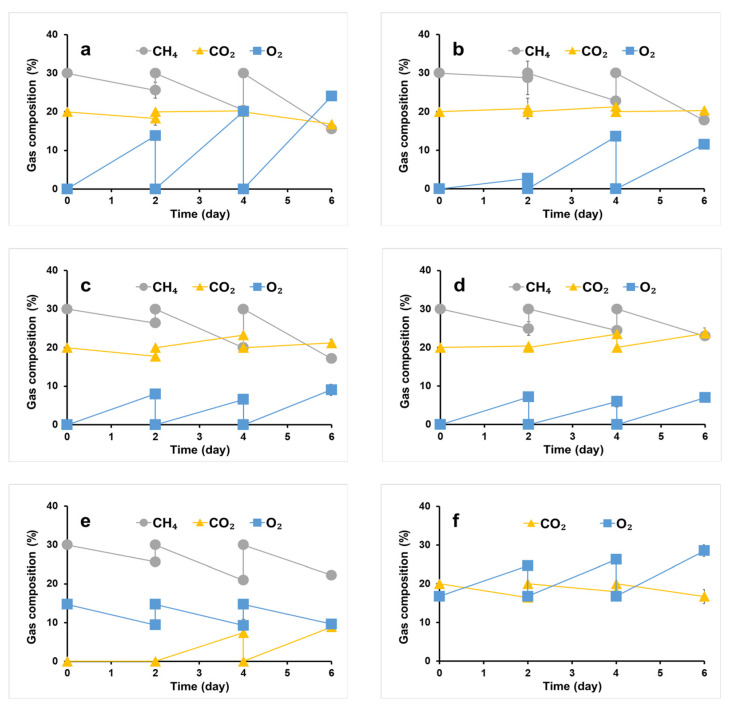
Gas consumption of syntrophic consortium of *M. trichosporium* OB3b with phototrophs in OPGs at 11.5 mM nitrogen (**a**), 5.7 mM nitrogen (**b**), 2.9 mM nitrogen (**c**), 1.4 mM nitrogen (**d**), *M. trichosporium* OB3b (control; 9.9 mM nitrogen) (**e**), OPGs (control; 1.6 mM nitrogen) (**f**).

**Table 1 microorganisms-11-01110-t001:** Biomass and PHB production from the syntrophic consortium of *M. trichosporium* OB3b and OPGs using different nitrogen source concentrations after 7 days of incubation.

Nitrogen Source Concentration (mM)	Biomass (g/L)	PHB Content (mg/100 mg Biomass)	PHB Titer (mg/L)
11.5	1.30 ± 0.03	2.6 ± 0.2	33.3 ± 2.1
5.7	1.29 ± 0.00	2.8 ± 0.4	36.6 ± 4.6
2.9	1.13 ± 0.04	7.4 ± 0.0	83.0 ± 2.6
1.4	0.50 ± 0.06	13.7 ± 1.5	67.3 ± 1.4
9.9 (OB3b control)	0.44 ± 0.01	38.1 ± 0.0	167.6 ± 5.4
1.6 (OPGs control)	0.53 ± 0.05	0.2 ± 0.0	1.1 ± 0.2

## Data Availability

Not applicable.

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
