# Peer review of "Polyhydroxybutyrate Production from Methane and Carbon Dioxide by a Syntrophic Consortium of Methanotrophs with Oxygenic Photogranules without an External Oxygen Supply"

_microorganisms, 2023, doi:10.3390/microorganisms11051110_

Round 1

Reviewer 1 Report

In this study, the authors described methane-CO2 mixed gas conversion into PHB by the bioprocess using both methanotrophic bacteria and oxygenic photosynthetic bacterial communities, and showed that this bioprocess is effective for the gas conversion. The authors experimentally demonstrated that the change in consumption of methane and CO2 in the bioprocess on the type of methanotrophic bacteria, and that M. trichosporium OB3b is suitable for the methane-CO2 mixed gas conversion process among the two species examined. Furthermore, the process was investigated using growth media with different concentrations of nitrogen source, and it was shown that PHB synthesis requires a relatively low concentration of nitrogen source. These results suggest a new process for converting methane and CO2, which are considered to be high contributors to global warming, into plastics, and are worthy of publication in the journal.

However, we suggest that some corrections and additions to the description and data are necessary to improve the validity of the description and the understanding of the readers of the journal. We recommend that the authors consider the following comments and revise the manuscript.

Comments

1) P1, Lines 28-31: The authors cite a paper (Reference 1) and the 2014 IPCC report (AR5) on the greenhouse effect factor of methane and methane emissions. The latest information on these was published in the IPCC AR6 in 2021. I recommend that the authors cite this latest report to confirm and update the values.

2) P5, Line 200-202: Here, the increase of O2 into the headspace and the decrease of CO2 are noted. I agree with the authors that this trend is very strong, as shown in Figure 1c, starting 4-5 days after the start of cultivation. However, during the first few days of culture, although methane is consumed, CO2 consumption and O2 accumulation are very slow. In other words, there is a clear difference in the gas composition of the headspace before and after the first 4-5 days-cultivation. Explanation regarding this point enhance the reader's understanding of the authors' experimental results.

Perhaps the reason for the smaller CO2 consumption immediately after the start of cultivation compared to methane consumption is due to the rate balance between CO2 production by methane metabolism of methanotrophs and CO2 consumption by OPGs, which prevents a significant decrease in the APPARENT CO2 content. I understand that this point is recognized by the authors because they describe it in Lines 206-207.

Also, the authors analyzed the gas components of the headspace in this experiment, and the solubility of carbon dioxide in the aqueous phase is much higher than that of methane and oxygen. I think that if the authors had considered this difference in solubility in your analysis and discussion, it would have deepened the discussion of the data. Please consider adding explanations regarding this point as well.

3) p6, Line 227-228: It is stated that high pH inhibited the growth of methanotrophic bacteria. I think that experimental data or examples of previous reports by the authors are needed on this point. We recommend adding more data or references.

4) The data of this study on pH change is very suggestive. Is there a possibility that this PHB synthesis system can be applied to Type I methane oxidizing bacteria by applying a pH controlled culture method? If the authors think it is likely, could you briefly touch on this point in the manuscript?

5) P8, Figure 3 & P9, Table 1: The culture conditions for Control, especially the concentration of the nitrogen source, are not clear; the authors need to be clearly stated in the Figure and Table captions.

6) P9, Table 1: I think the authors were comparing the amount of PHB synthesis at the same cultivation time (described as 7 days in Materials & Methods). I suggest adding the cultivation period in the caption of Table 1 to make this clear.

Reviewer 2 Report

The results and discussion provided by the authors are well presented.

Therefore, I recommend publishing this paper in its present form.

Author Response

We are deeply grateful to the reviewer for the comprehensive analysis of our manuscript and giving positive feedback.

Reviewer 3 Report

The manuscript is nicely written, and including 16S rRNA amplicon experiments brings some fresh perspective to similar studies studied previously. Yet, there are some suggestions to improve the manuscript further. 

1. Line 33: Complete the sentence: Developing efficient processes to convert CH4 into “useful products/value-added products” is crucial.

2. Line 92: Provide a reference for nitrate mineral salts (NMS) medium. 

3. Line 122: Correct the title to 16S rRNA amplicon sequencing.

4.   Line 182: Both conditions had a CH4 to CO2 ratio of 3:2, representing a typical biogas composition. Please provide a reference here. 

5. Materials and Methods: Authors should be more detailed with their materials and methods to enable clear protocols to the reader or provide references—example 2.2. Write the inoculum % of cultures used—autoclaving media conditions, what components were added before autoclaving and what after, etc. For readers unfamiliar with methanotrophs and oxygenic photo granules (OPGs), it would be nice if authors could write what conditions were used to maintain these stocks/cultures, etc. 

6.   Following the general convention, authors should write down the model no/companies of instruments used, wherever mentioned, for example, which of the gas controller system was used, etc. 

7. Materials and Methods: PHA extraction: Line 170: Provide a reference or expand the extraction procedure to add relevant details. 

8. Line 219: Can authors think of a reason for the low growth of Methylomonas sp. DH-1 compared to M. trichosporium OB3b under carbon lean conditions? Please elaborate. 

9. Line 225: Can authors think of a reason for the increased pH in cultures during carbon-limited conditions compared to carbon-rich conditions?

10. A clear section of the hypothesis needs to be included in the introduction.

11. Section 3.2: The number of samples analyzed needs to be clarified. How many samples, and how many replicates? How many DNA extractions and sequencing?

12. Throughout the manuscript: Did the author use the triplicate sample for the analysis? Please include some statistical portions of this study.

13. Throughout this study, especially in section 3.3, the authors provide a possibility of active growth of Methylomonas sp. DH-1 or M. trichosporium OB3b relative to or depending on how the consumption profiles were for methane. In such studies involving methanotrophs in consortia, researchers typically adopt FACS or FISH with specialized tags to quantify the abundance of methanotrophic strain under study. Adoption of any such strategy in this study would have been helpful too.

14. The PHB titers obtainable in this study, the highest being 83 mg/L in the consortium, is much less. Can the authors discuss here the advantage of adopting such a consortium of methanotrophs with OPGs, which do no good for PHB yields overall? 

15. Authors are recommended to include some qualitative results for PHA extracted, at minimum, an FTIR graph and some measurements for the purity of the PHB. This is recommended because extraction of PHB from consortium is troublesome and hinders extraction of PHB in its purified state to be accurately quantified by GC.

16. Discussion needs to be included in terms of comparison of results and observations in this study with previously published papers corroborating similar consortia for PHB production. Authors are advised to strengthen their discussion in this direction. 

17. Authors mention presence of Pseudomonas sp. in the OPG’s consortia. Typically, Pseudomonas sp. are notorious for having a conserved system for synthesis of mcl-PHA. In this study, there is no account from the authors with regards to the possibility of PHA being produced in addition to PHB, in the consortium under study. Please elaborate.

18. It will be nice if the authors point out towards the novelty of this study in its introduction, compared to previously existing research papers in the introduction. 

Round 2

Reviewer 3 Report

Thank you for addressing my suggested comments. I have no further suggestions.